

**Wet-Radome Attenuation in ARM Cloud Radars and Its Utilization in Radar Calibration Using**
**Disdrometer Measurements**
Min Deng[1], Scott E. Giangrande[1], Michael P. Jensen[1], Karen Johnson[1], Christopher R. Williams[2],
Jennifer M. Comstock[3], Ya-Chien Feng[3], Alyssa Matthews[3], Iosif A. Lindenmaier[3], Timothy G.
Wendler[3], Marquette Rocque[3], Aifang Zhou[1], Zeen Zhu[1], Edward Luke[1], and Die Wang[1]
[1] Brookhaven National Laboratory, Environmental and Climate Sciences Department, Upton,
New York
[2] University of Colorado Boulder, Colorado Center for Astrodynamics Research, Boulder,
Colorado
[3] Pacific Northwest National Laboratory, Richland, Washington
*Correspondence to*: Min Deng (mdeng@bnl.gov)
Manuscript to be submitted to AMT publication.





**Abstract**

A relative calibration technique is developed for the U.S. Department of Energy's (DOE) Atmospheric Radiation Measurement (ARM) user facility Ka-Band ARM Zenith Radars (KAZRs). The technique utilizes the signal attenuation due to water collected on the radome for estimates of the reflectivity factor (Ze) offset. The wet-radome attenuation (WRA) is assumed to follow a logarithmic relationship with rainfall rate in light and moderate rain conditions, measured by a collocated surface disdrometer. A practical advantage of this WRA approach to shorter-wavelength radar monitoring is that while it requires a reference disdrometer, it is shown viable for a wider range of collocated disdrometer measurements than traditional disdrometer direct comparisons in light rain. Adding such techniques may provide an additional, cost-effective monitoring tool for remote/longer-term deployments.

This technique has been applied during the ARM TRacking Aerosol Convection interactions ExpeRiment (TRACER) from October 2021 through September 2022. The estimated offsets in Ze are evaluated against traditional radar calibration and monitoring methods based on datasets available during this campaign. This WRA technique reports offsets that compare favorably with the mean offsets found between the cloud radars and a nearby disdrometer near the time of rain onset, while also demonstrates similar offset and campaign-long trends with respect to collocated and independently-calibrated reference radars. Overall, the KAZR Ze offsets estimated during TRACER remains stable and at a level 2 dBZ lower than the Ze estimated by disdrometer from the campaign start until the end of June 2022. Thereafter, the radar offsets increase to near 7 dBZ at the end of the campaign.




**Short Summary**

A relative calibration technique is developed for the cloud radar by monitoring the intercept

of the wet-radome attenuation (WRA) logarithmic behavior as a function of rainfall rates in light
and moderate rain conditions. This WRA technique is applied to the measurements during the
ARM TRACER campaign and reports Ze offsets that compare favorably with the traditional
disdrometer comparison near the time of rain onset, while also demonstrates similar offset and
campaign-long trends with respect to collocated and independently-calibrated reference radars.




**1 Introduction**

The U.S. Department of Energy (DOE) Atmospheric Radiation Measurement (ARM) user

facility operates multiple millimeter-wavelength cloud radars (at 35 and 94 GHz frequencies)
across a variety of global fixed and mobile facilities (e.g., Mather and Voyles, 2013; Miller et al.
2016; Kollias et al., 2007; 2020). The popularization of "cloud" radars for use in atmospheric
research is tied to the fact that they are often more sensitive than conventional weather (i.e., cm
wavelength) radars for detecting cloud droplets. One trade-off for these radars is that they
experience partial attenuation to potential extinction in clouds and precipitation. Importantly, key
quantitative cloud property and hydrological retrievals from cloud radars often carry an uncertainty
that is tied to the accuracy to which its quantities (such as the radar reflectivity factor Ze) can be
estimated in the presence of attenuation in the atmosphere (Matrosov, 2005; Meagher et al., 2006;
Deng et al., 2014; Zhu et al., 2019, Liu et al., 2022).

Given the importance of accurate Ze measurements, the routine deployment and operation

of cloud radars necessitates frequent calibration and monitoring activities. In general, more
rigorous radar calibration efforts can be operationalized (e.g., Russchenberg et al., 2020), but these
approaches are often system-specific and require highly skilled engineers or technicians,
significant time and specialized equipment (within ARM, i.e., Mead 2010). For weather and
climate applications, radar-based research has often turned to an increasing number of "relative"



calibration techniques that include concepts that rely on estimates of Ze from nearby reference
instrumentation, or expectations based on intrinsic properties of the hydrometeors or other media
(e.g., Bringi and Chandrasekar 2001; Giangrande et al. 2005; Protat et al., 2011; Kollias et al.,
2019; Maahn et al., 2019; Williams et al., 2023). Several such "natural" calibration concepts have
proven effective for quantifying radar performance for many hydrological applications that require
Ze estimates to within 2-3 dBZ. Yet, the simplest approach is often to perform a cross-comparison
of Ze characteristics to collocated and (assumed) calibrated reference radars. For example,
extended comparisons using clouds near ARM ground sites using CloudSat radar measurements
has been successful for the monitoring of the long-term ARM cloud radar record (Protat et al.,
2011; Kollias et al., 2019). For finer-scale comparisons during ARM deployments, the Ka-Band
ARM Zenith Radar (KAZR) is often collocated with a Radar Wind Profiler (RWP, 915 or 1290
MHz) and the Ka- and X-band Scanning ARM Cloud Radar (SACR, or KaSACR and XSACR)
that are easier to monitor using independent techniques better suited to scanning and/or longer-
wavelength radar.
Among the many forms of relative cloud radar monitoring, a common method relies on
surface disdrometer observations. Reflectivity factor can be estimated for assumed rain properties
using techniques such as T-matrix scattering algorithms applied to the surface disdrometer-
measured drop size distribution of rain (Mishchenko et al., 1996). The comparison of radar-
measured reflectivity ($Ze_{meas}$) near the surface disdrometer-estimated ($Ze_{dis}$) provides one common
path to estimate radar calibration offsets (e.g., Kollias et al., 2019, Myagkov et al., 2020;
Russchenberg et al 2020 and Lamer et al 2021). Disdrometer comparison techniques of this sort
have been implemented as routine procedures for radar monitoring, such as for the Aerosol Cloud
Tracer Gas Research Infrastructure (ACTRIS) network in Europe (Dupont et al, 2022). For radars
that experience negligible attenuation in rain, such procedures are often straightforward to
implement under a variety of widespread precipitating conditions (e.g., Williams et al., 2023).
However, for shorter radar wavelengths where gaseous attenuation, attenuation in rain, and wet-
radome attenuation are not negligible, the application of this idea can become more complicated.
Specifically, the two-way attenuation associated with radome wetting (i.e., the wet radome
attenuation or WRA herein), is a well-known phenomenon. During rainfall, water droplets bead
on the surface of the radar radome, and this rain may form a wet film that eventually flows off the
radome when this film achieves sufficient mass. Droplets impacting this radome during persistent



rain further alter the water depth on the radome through bouncing and splashing (Gibble 1964,
Anderson 1975, and Yu et al. 2021). For long wavelength radars, this WRA is often considered to
be negligible (Thompson et al. 2012, Kurri and Huuskonen 2008). For shorter wavelength radars,
the impact of WRA is potentially more significant. For example, at X-band, Bechini et al. (2010)
and Gorgucci et al. (2013) found a loss of 5 dB in moderate rain through comparison of
simultaneous X-band radar measurements at close range with a collocated video disdrometer. This
WRA has been shown to depend on the thickness of the water film (*d*) on the radome, which in
turn is a function of rain rate through the Gibble formula (Gibble 1964, and Anderson 1975):
$$d = \left(\frac{3\mu_k rR}{2g}\right)^{1/3} \qquad , \qquad (1)$$

where $\mu_k$ is the kinematic viscosity of water (that also varies with temperature), *r* is the radome
radius, *R* is the rain rate, and *g* is the gravitational acceleration. Additional relations between WRA
and *R* have been developed based on the Gibble's $R^{1/3}$ formula by Frasier et al. (2013) and
Gorgucci et al. (2013) for X-band radar calibration studies.

Few studies have considered WRA for assessing cloud radar offsets at Ka-band (35 GHz).

At this frequency, one expects a stronger two-way attenuation for the same depth of rainwater on
the radome, as the water absorption coefficient is approximately three times larger than at X-band
(Bertie et al. 1996). It is understood that WRA will impact direct estimates of the offset between
cloud radar and disdrometer Ze estimates in rainy conditions, and faulty offset assessment after
rain ends may occur owing to extended radome drying delays. Therefore, direct comparison
concepts previously cited typically consider only the periphery cloud, drizzle or light rain
conditions (i.e., $R < 1\text{-}2$ mm hr$^{-1}$) at the onset of a rainfall event to minimize various forms of
attenuation. This often is a very stringent and subjective employment of these conditions: First, it
limits the opportunities for direct disdrometer monitoring of cloud radar to a selected window of
rainfall rates and event timing. Identifying these light rain or drizzling conditions is also contingent
on the requirements for collecting high-quality disdrometer measurements (i.e., those that require
significant droplet number counts), wherein a separate rain rate cut-off may be required to avoid
significant WRA. Overall, it is potentially useful to establish other forms of cloud radar monitoring
that could benefit from a wider range of observations collected during precipitation window.



In this study, we identify intervals of WRA for Ka-band radars by comparing observations
from ARM's KAZR and a collocated suite of instruments including surface disdrometer, calibrated
RWP and SACR observations collected in vertical pointing (VPT) modes during the Tracking
Aerosol Convection interactions ExpeRiment (TRACER). The KaSACR and XSACR radar
observations benefit from the radars' ability to shed radome water during scanning, therefore less
influenced by WRA. Section 2 introduces the radar datasets and the supporting TRACER datasets
used in this study. In Section 3, by implementing a logarithmic relation between WRA and rain
rate in light to moderate rain, a relative calibration technique is developed. This technique monitors
the intercept of this logarithmic relationship for daily KAZR measurements collected during WRA
conditions into moderate rainfall cases. In Section 4, the technique is applied to the daily KAZR
measurements during the TRACER campaign to assess the KAZR long-term calibration offset
trend. The performance of this technique is evaluated against three traditional relative calibration
or monitoring methods for Ka-band radar: (i) the direct disdrometer comparisons of Ze in light
rain at the onset of rain events, (ii) a cross-comparison with independently-calibrated RWP
measurement, and (iii) a cross-comparison with collocated scanning KaSACR measurement. A
summary of the performance for these WRA techniques for relative offset monitoring is found in
Section 5.

**2 TRACER Dataset Description and Comparisons**
The TRACER campaign took place in the Houston, TX region from 1 October 2021 to 30
September 2022 (Jensen et al., 2019, 2022, and 2023) with a goal of studying the interactions of
aerosols and convective clouds. The main surface measurement site was located at La Porte, TX
(29° 40' 12'' N, 95° 3' 32.4'' W) that housed the deployment of the first ARM Mobile Facility
(AMF1; Miller et al., 2016). The AMF1 consists of several ground-based remote-sensing and
profiling instruments, and included the deployment of the KAZR, KaSACR, XSACR, and RWP
units that serve as the radars for this study. The surface instrumentation also included multiple
laser and video disdrometers as reference anchors.

**2.1 TRACER Cloud Radars (KAZR and SACR)**
The KAZR (Widener et al., 2012) is a follow-on to ARM's widely successful millimeter-
wavelength cloud radar (MMCR). The KAZR has a flat radome, inclined at 4°. A complete listing





of KAZR specifications is found in Table 1. The KAZR transmits and receives two types of pulses:
(i) the burst pulse, which is a simple narrow pulse of radio-frequency energy (referred as "GE"
mode), and (ii) the chirp pulse, which is a longer, frequency-modulated pulse with higher
transmitted energy and higher sensitivity, but with data collection starting at a higher range due to
the larger blind zone imposed by the longer pulse length (referred as "MD" mode). Though the
MD mode is more sensitive to clouds (i.e., lower minimum detectable Ze), only the KAZR GE
mode data are used for disdrometer comparisons since near-surface observations are needed.

The KaSACR and XSACR are co-mounted on a scanning pedestal (SACR, e.g., Kollias et

al., 2014a and 2014b). During TRACER, the KaSACR and XSACR nominally repeated a 10-
minute scanning pattern: (i) two low-level plan position indicator (PPI) scans at 1° and 2°
elevation, followed by, (ii) 6 hemispheric range height indicator (HSRHI) scans at 30° azimuth
intervals, then (iii) 2 minutes of VPT mode. This study draws the 2-minute VPT mode  from its
10-minute scanning scanning sequence (i.e. nominal scanning VPT mode). The specifications for
the SACR during VPT modes are listed in Table 1. For one event during the campaign (03-04
September 2022), the SACR was temporarily operated in an exclusively VPT mode (i.e. stationary
VPT mode) for  the radar cross-calibration purposes. The KaSACR has an inclined radome similar
to the KAZR, but is relatively newer (i.e., less potential deterioration of its hydrophobic coating).
The XSACR has a conical radome with a slant angle of 45$^o$ to the surface. Overall, the WRA effect
should be smaller for the XSACR than either Ka-band radars due to known wavelength
dependency differences, as well as this improved radome design. The KaSACR calibration offsets
between May and September 2022 are expected to be stable based on the ground clutter analysis
with the relative calibration adjustment (RCA) techniques (Skolnik 2000 and Hunzinger et al.,
2020) and are close to 0 dB according to the ARM TRACER radar b1 data processing report (Feng
et al. 2024). To be compared with $Ze$ estimates from VDISQUANTS, radar measurements at 500
m are selected and corrected for gaseous attenuation using nearby radiosonde measurements (e.g.,
Ulaby et al., 1981). The rain attenuation is also corrected from specific attenuation coefficient ($K$)
estimates from VDISQUANTS, assuming a uniform layer between surface and 500 m.

There is a concern that the radar might be saturated, especially for the KaSACR near at its

minimum range, which could cause low bias in the measured Ze compared to disdrometer Ze.
Based on a communication with ARM radar engineer, the power associated with the highest
voltage digitizable by the radar's Analogue-to-Digital Converter (ADC) is 5.9 dBm. The





corresponding KAZR saturation reflectivity at 500 meters is about ~45 dB with its calibration
constant of -12 dBm. Similarly, the saturation reflectivity at 500 m is about ~31 dB for KaSACR,
with its calibration constant of -26 dBm. While the measured radar reflectivities from both KAZR
and KaSACR at 500 m are generally less than 25 dBZ, well below saturation. Further supporting
proof through the comparison of radar profiles can be found in Supplement material.

**2.2 Surface Disdrometer Measurements and Value-Added Products**

A Parsivel2 laser disdrometer (LDIS) and a two-dimensional video disdrometer (VDIS)

unit were deployed at the main site during TRACER in very close proximity to the cloud radars.
For disdrometer geophysical quantities and data quality control, procedures follow the standard
drop size distribution (DSD) filtering in Giangrande et al. (2019) implemented by ARM in their
precipitation value-added products (Video Disdrometer Quantities--VDISQUANTS and Laser
Disdrometer Quantities--LDISQUANTS, Hardin et al., 2020). These products employ several fall
speed checks, temperature, drop shape/canting assumptions, larger drop restrictions (no drop sizes
> 5 mm) and drop count thresholds (> 20 drops per minute for a valid DSD) that impact estimates
of hydrometeor Ze and rain-specific attenuation coefficient ($K$) for radar frequencies using a T-
matrix scattering algorithm (Mishchenko et al., 1996). As further discussed within the disdrometer
literature (Tokay et al., 2001, 2013; Giangrande et al., 2019; Wang et al., 2021), the VDIS is
considered the more reliable and sensitive disdrometer to a wider range of drop sizes under
nominal light rain operating conditions. Therefore, the estimated Ze at Ka-band in VDISQUANTS
is used within this study as our ground truth for KAZR calibration and surface rain rate, while the
LDIS products have been used as an independent reference for monitoring RWP Ze estimates (e.g.,
Williams et al., 2023), which is required for additional direct radar comparisons in Section 4.

**2.3 Radar Wind Profiler (RWP)**

The RWP deployed during TRACER was operated using an adaptive scanning mode,

switching between a traditional boundary layer horizontal wind mode and a vertically-pointing
precipitation mode adopted by ARM for its recent deep convective cloud campaigns (e.g., Tridon
et al., 2013; Giangrande et al., 2013, 2016).  When the signal-to-noise ratio in the vertical beam
exceeded a predefined threshold, the RWP switched into this precipitation mode and employs a
single vertically-pointing beam operation. This mode transmitted short- and long-pulses to observe



echoes close to the radar with fine resolution, or further from the radar with coarser
resolution. Important to this study, the TRACER RWP mode switching sometimes prevented the
RWP from immediately observing the periphery lightly precipitating clouds as they passed over
the AMF1 site. However, this mode-switching sampling issue does not impact the bulk KAZR-
RWP Ze cross-comparisons because we primarily consider daily average behaviors. As before, the
RWP Ze measurements in precipitation mode were calibrated independently using collocated
LDIS observations (i.e., Williams et al., 2023), who found a standard deviation of 2 - 4 dBZ
between the RWP at 500 m and LDIS.
**3 Cloud Radar Ze Calibration and Monitoring: Development of a New WRA Technique**
**3.1 Identification of WRA: SACR in Stationary VPT Modes**
Figure 1a-c show the measured Ze from the XSACR, KaSACR, and the KAZR GE mode
on 03-04 September 2022, when SACR was operated exclusively in a stationary VPT mode. Two
rain intervals were captured with widespread rainfall, with the first around 17-19 UTC, and the
second from 20 – 02 UTC. A radar "bright band" signature around the meting level
(approximately, 5 km AGL) is observed for this event. After 02UTC (20 LT), light rain was
followed by scattering high clouds in the overnight period until thick anvil clouds from other
nearby convection moved in (15 UTC, 09 LT). Overall, the XSACR and KaSACR report similar
Ze values in the periphery cloudy conditions and for initial samples in light rain when attenuation
in rain and WRA should be minimal. Expectedly, the larger discrepancies between XSACR and
KaSACR (with the KaSACR reporting lower, attenuated Ze) are found during the relatively
heavier rainfall period between 2200-0000 UTC. The KAZR Ze is consistently reporting lower
values than those from the KaSACR, with this difference often exceeding 5 dBZ throughout the
event.
The Ze difference between the KaSACR and KAZR values in Fig.1d. shows a strong
temporal variation, but limited vertical variation, indicating that the difference is not driven by
atmospheric features, but by the radar or its near environment, such as the WRA. The minimum
difference between the radars is ~7 dB is found in high clouds around 17-18 UTC and the next
morning (15-17UTC) on 4 September, a strong indication of the overall Ze offset between KAZR



and KaSACR. The minimum difference of ~7 dB in rain (19, 21 and 23 UTC) indicates that WRA
for KAZR and KaSACR behavior similarly.

An increased and prolonged difference after moderate rain, especially for the humid

environment at night (0-12UTC, or 18-6 LT) indicates that KAZR and KaSACR carry additional
sources of discrepancy after rain ends or under high humidity since the KAZR radome is older and
less hydrophobic than the KaSACR radome, as argued in radar calibration during the Cloud,
Aerosol and Complex Terrain Interactions campaign (CACTI; Varble et al. 2021; Hardin et al.
2020). Accurate correction for KAZR wet-radome attenuation is very challenging and beyond the
scope of this study, however the WRA behavior in rain can be used to track KAZR calibration, as
will be demonstrated in the following.

The time series of rain rate ($R$), $K$ and $Ze$ estimates at Ka- and X-bands from

VDISQUANTS for the 03-04 September 2022 case are shown in Fig. 2a and b. The sampled $R$
from the disdrometer are commonly less than 1 mm hr$^{-1}$, but approach 5 mm hr$^{-1}$ around 2330
UTC. The Ze from KAZR, KaSACR, and XSACR at 500 m are plotted in Fig, 2b. For all
collocated precipitating samples, the XSACR Ze (black crosses) has a high correlation with
estimated Ze (rr = 0.95), while KAZR Ze (blue crosses) are biased low when directly compared to
the disdrometer Ze, which is exacerbated further in heavy rain contexts. KaSACR Ze (red cross)
falls in between XSACR and KAZR Ze.

Fig. 2c shows the differences between measured and estimated Ze ($Dze$) for KAZR,

KaSACR and XSACR. The XSACR has a minimum $Dze$ of 0 dB when the rain rate is less than
0.1 mm hr$^{-1}$, but this can be as large as 5 dB around 23:30 UTC. The KaSACR $Dze$ is
approximately 1 dB at 18 and 21 UTC, while the KAZR $Dze$ is around 7 dB (possibly indicating
that KaSACR and KAZR calibration offsets are near 1 and 7 dB, respectively). Both KaSACR and
KAZR Ze are further biased lower by another 13 dB when rain rate is close to 5 mm h$^{-1}$ around
23:30 UTC. This 13 dB decrease in KAZR and KaSACR estimates is substantially larger than the
two-way attenuation in rain droplets at Ka-band (~2 dB, Fig. 2a), suggesting that other factors such
as WRA are increasingly contributing to the offset in rain and WRA for both KAZR and KaSACR
likely exhibits a similar rain rate dependence.





The estimated Ze from VDISQUANTS during the entire TRACER campaign are plotted
as a function of $R$ in Fig. 3. The estimated Ze for both X- and Ka-band has a log-linear dependence
of $R$. When $R$ is larger than 2 mm hr$^{-1}$, the Ze values diverge and the difference between the two
wavelengths increases as the $R$ increases due to the resonance effects of non-Rayleigh scattering
(Baldini et al., 2012). The cumulative probability distribution (CDF) of rain rates (red line in Fig.
3) shows that the percentage of disdrometer data samples with $R < 0.1$ mm h$^{-1}$ are ~15%, indicating
few samples for the application of traditional, direct disdrometer comparison at precipitation onset.
However, approximate 85 % of TRACER data samples suggest $R < 5$ mm h$^{-1}$, which may be
suitable for the WRA technique applications to follow.
**3.2 Identification of WRA: SACR in its Scanning-VPT Mode**
To further illustrate the WRA, we compared radar and disdrometer measurements while
SACR was operating in its nominal 10-minute scanning sequence in a stratiform rain event
observed on 11 August 2022, between the hours of 01 - 04 UTC (Fig. 4). The radars were under
persistent rain, ranging from 1 mm hr$^{-1}$ at 01 UTC to more than 5 mm hr$^{-1}$ around 02:15 UTC
which caused strong attenuation of the radar signal, especially visible in the KAZR Ze vertical
gradient above 4 km (Fig 4a). After 03 UTC, the rain at the surface was so light that the disdrometer
was unable to measure rain DSDs effectively for Ze estimates due to too few drops (< 20/minute)
(Fig. 4b).
The surface disdrometer-estimated Ze at Ka- (black diamonds) and X-bands (blue
diamonds) shown in Fig. 4c are all close to 30 dB when the rain rates are near 1 mm hr$^{-1}$, while
the KAZR Ze is near 15 dB, resulting in a $Dze$ against disdromter of 15 dB, as plotted in Fig. 4d.
As the SACR was operating in its nominal scanning pattern during this event, there is an 8-minute
gap in measurements associated with the PPI and HSRHI scanning sequences for every 2 minutes
of VPT measurements. The collocation of the 2-minute VPT data is extended to 6-minute of data
with a ±2-minute averaging window between SACR and VDISQUANTS.
The KaSACR Ze values (red cross) in Fig. 4c display 6-minute sawtooth behaviors in every
10-minute scanning heartbeat. This pattern starts with values closer to XSACR Ze at the beginning
of each sawtooth, then it decreases towards the KAZR Ze value as time increases, with scaling
potentially correlated with the rain rate. In contrast, the 03-04 September 2022 case in Fig. 2b



shows parallel Ze trends between KAZR and KaSACR. The increasing *Dze* trend in every 6-
minute measurement (red cross) in Fig 4d is more apparent. The sawtooth behaviors of Ze or Dze
in KaSACR in this case illustrate that the extra Ze bias is caused by increasing rain accumulation
on the radome during the 2 minutes of vertical pointing.  If, on the other hand, the radar signal
were saturated, it would be saturated all the time rather than bouncing back and forth. A closer
examination of XSACR *Dze* trend (black cross) in Fig. 4c and d, reveals very little consistent
variability with rain rates in the scanning cycle, likely owing to a weaker water absorption
coefficient at X band and less water collecting on the conical radome of XSACR.
The differing KaSACR patterns between events from Figures 2 and 4 are related to
rainwater accumulation and SACR radar cycling between the scanning and stationary VPT modes.
At the beginning of the scanning VPT period, the radome is covered with a relatively thin film of
rainwater since the radome shed the water during the RHI and PPI scanning. Excess rainwater
quickly accumulates on the radome in the VPT mode, causing enhanced attenuation. Therefore,
the WRA for the KaSACR is modulated by the 10-minute scanning cycle. Alternatively, for
observations of KAZR and KaSACR in its stationary VPT mode on 03-04 September, rainwater
accumulated on their radomes in a consistent/continuous way, therefore the WRA patterns are
similar and the measured Ze and *Dze* are parallel to each other with a constant offset of about 7
dB.

**3.3 WRA Fitting Calibration Technique**
In this section, we examine the WRA behavior toward developing a relative calibration
technique for cloud radar monitoring. Figure 5a shows the estimated Ze (black cross) by KaSACR
at 500 meter after gaseous and rain attenuation corrections and the corresponding VDISQUANTS-
estimated Ze (red cross) as a function of rain rates for the 03-04 September case. A very well-
correlated monotonic relationship between the VDISQUANT-estimated Ze and $R$ in logarithmic
space is observed. However, the KaSACR-measured Ze is biased low compared to the estimated
Ze, and the offset between them ($D_{Ze} = Ze_{\text{dis}} - Ze_{\text{meas}}$ shown in Fig. 5b) increases as $R$
increases. The *Dze* is near 0 dB at $R < 0.1$ mm hr$^{-1}$, when water films may not form on the radome
– thus, minimal WRA is expected. *Dze* increases up to 15 dB at $R \sim 5$ mm hr$^{-1}$. The WRA with
magnitude up to 15 dB is potentially a disadvantage when considering cloud radar observations in



precipitation. However, this magnitude and range of attenuation as a function of $R$ provides a
unique opportunity to explore relative radar calibration techniques.

Given a quasi-linear correlation between $Dze$ and $R$ in logarithmic space in Fig. 5b, we can

perform a weighted linear least-squares fitting of the $Dze$ with $R$ in logarithm in the following Eq.

2:


$$D_{ze} = a + b \log(R) \tag{2}$$

For the cases in Fig. 5b, the fitted slope $b$ are estimated to be 8.6.  The intercept "$a$"

captures the radar calibration offset and the WRA when $R$ is 1 mm hr$^{-1}$. As the KaSACR calibration
offset is close to 0 then the intercept due to the WRA effect with $R$ equal to 1 mm hr$^{-1}$ is around
11.1 dB.

This log-linear relation between $Dze$ and $R$ is different from the Gibble's formula of $R^{1/3}$

(Eq.1) applied by Frasier et al. (2013) and Gorgucci et al. (2013) for X-band radar calibrations. As
the water absorption coefficient at Ka-band is about three times that at X-band, we divide the Eq.
2 of the log-linear fitting result by 3 and plot it with the fitting relations in Frasier et al. (2013;
solid blue line) and Gorgucci et al. (2013; solid black line) in Figure 6. We find that the relationship
derived in this study intersects with those of Frasier et al. (2013) and Gorgucci et al. (2013) at $R$
of 0.2 mm hr$^{-1}$, where the majority of the data from our study are concentrated. When $R$ > 0.2 mm
hr$^{-1}$, this WRA fitting result is larger than Gorgucci et al. (2013) by less than 0.5 dB, although the
Gorgucci et al. (2013) behavior is larger than Frasier et al. (2013) by 0.5 - 1 dB. When $R$ < 0.2 mm
hr$^{-1}$, our WRA fitting result is smaller than the others two by about 0.5 - 1 dB. This difference
between this log-linear fitting and previous studies (1 dB) is smaller than the data scatter found in
Fig. 5b (with a standard deviation of 3 dB), and smaller than the difference between the two
previous studies. This potentially indicates that the log-linear fitting function in Eq. 2 is reasonable
for WRA correction when $R$ *is* less than 5 mm hr $^{-1}$, previously selected as the threshold for our
data of interest.

As the radar calibration offsets are assumed independent of $R$, and the WRA has an intrinsic

characteristic dependence of $R$, then the radar calibration offset can be obtained by monitoring the
fitted intercept in Eq. 2. Fig 5e illustrated the intercept offset of the fitted $Dze$ *lines* between the
KAZR(red cross)  and KaSACR (dashed black line). The fitted intercept of KAZR is 18.5 dB,





about 7.5 dB higher than that of KaSACR, which is consistent to the offset between KaSACR and
KAZR we observed from comparisons in Figure 1d and the time series in Figure 2c.

On the other hand, we can also assume negligible WRA when $R$ is small, e.g., $R = 0.05$

mm hr$^{-1}$, then the *Dze (R = 0.05)* is the radar calibration offset, which can be used for the radar
operation monitor. For the KaSACR on 03-04 September case in Fig 5a, the *Dze (R= 0.05)* is -0.1
dB, while for the KAZR, the *Dze (R= 0.05)* is 7.3 dB, which is consistent with direct KaSACR
and KAZR comparison and their comparison with VDISQUANTS. This suggests that the WRA
technique provides reliable offset estimates for this case. The corrected Ze with the log-linear fitted
*Dze* in Eq. 2 are compared with the VDISQUANTS Ze in Fig. 5c and f for KaSACR and KAZR,
respectively. The correlation coefficient (*rr*) increases to ~0.9, the mean bias for both KaSACR
and KAZR is 0 dB and standard deviation is 3.0 dB.

To further explore the intrinsic WRA dependence on $R,$ we can apply this WRA linear

fitting calibration technique to KaSACR in its scanning-VPT modes. Due to water shedding in the
scanning cycle, we use the last-minute measurement of every 2-minute VPT period in the 10
minutes scanning heartbeat. To provide a variety of samples, we identified 5 stratiform rainy days
observed on May 25, August 05, 11, 19 and 29 and combined these events together. The collected
data from those 5 days are plotted along with the corresponding VDISQUANTS-estimated Ze (red
cross) as a function of rain rates in Fig. 5g. For these events, *Dze* $_{(R=0.05)}$ is -0.9 dB, with slope "b"
fit to 8.6. The adjusted Ze using this log-linear fitted *Dze* is compared with the VDISQUANTS Ze
in Fig. 7i is found to be well-correlated with the reference Ze with smaller standard deviation (rr=
0.91, 0 dB mean bias, and 2.0 dB standard deviation).   Recall the *Dze* $_{(R=0.05)}$ in stationary VPT
mode in 03-04 September case is -0.1 dB, the difference between the two KaSACR offsets is less
than 1 dB, which is well within the standard deviation of the estimated Ze (3 dB) as a function of
$R$, and is close to the 1 dB offset from the direct disdrometer comparison at light rain onset in Fig.
2. This suggests that the $R$ dependence of WRA is a valid assumption, therefore the interceptor or
*Dze* $_{(R=0.05)}$ in fitting Eq. 2 can useful for radar offset monitoring.

The time and height plots of Ze from KaSACR, XSACR, and KAZR GE and MD modes

on 03-04 September 2022 (after the WRA correction is applied) are shown in Figure 7. For the
precipitating period, KaSACR is adjusted with Eq. 2 appying a slope of 8.6 and constant of 11.1





(Table 2 or Fig. 5b). XSACR is modified with the offset of 3 dB from VDISQUANTs (black cross
in Fig 2d), and KAZR GE mode is corrected using Eq. 2 with a slope of 8.6 and an intercept
constant of 18.5 (Table 2, or Fig. 5b). For non-precipitating periods, the calibration offsets for
KaSACR and XSACR are assumed to be 0 dB based on the previous discussion, while the KAZR
GE mode is calibrated with an offset of 7 dB.  Compared to the apparent difference of more than
5 dB between KAZR and KaSACR suggested in Figure 1, the corrected Ze from KAZR and
KaSACR are similar to  those from XSACR in clouds and light rain. Under the relatively heavy
rain conditions (see,  2330 UTC), Ze in XSACR along the fall streaks retains magnitudes ~30 dBZ
from the surface up to the melting layer, while Ze estimates from KAZR and KaSACR gradually
decrease from the surface to the melting layer, presumably due to accumulating attenuation in rain
in Ka-band observations. This comparison in Figure 7 further supports the idea that the WRA
fitting technique can be applied to KAZR measurements and KaSACR in VPT modes, and can
provide reasonable estimates for wet-radome corrections in rain or radar offsets.

**4 Application and Evaluation of the WRA Offset Monitoring During TRACER**

**4.1 Daily TRACER KAZR Calibration Offset Applications**


We perform the WRA fitting technique on the *Dze* and *R* relationship using

VDISQUANTS Ze estimates versus KAZR Ze for each day with measured precipitation over the
entire TRACER campaign. The fitted slopes from the daily events typically range from 6 to 10,
with *rr* typically larger than 0.7. The fitted slopes and associated fitting errors depend on the data
sample distribution. For example, for rain events with short durations or limited intensity
variability, the data samples may cluster in a narrower range, thus the fitted Ze may suggest a
relatively lower correlation coefficient with the disdrometer Ze and be considered less reliable.

To avoid uncertainty associated with "daily" fitting as above (and/or lack of sampling

therein associated with additional daily spread), one may assume that the *Dze* and *R* relation has a
constant slope over longer windows. Here we consider applying the WRA fitting technique with
an average slope of 8, as a value selected to be representative for extended rain conditions over the
entire TRACER campaign dataset. As a sensitivity study of this composite slope choice, we
perform these offset calculations with proxy slope values at 6, 8 and 10 for both KAZR and
KaSACR on the 03-04 September 2022 case. The results for these tests are shown in Table 2. As
the slopes increase from 6 to 10, both the KAZR and KaSACR calibration offsets decrease by



about 3 dB, as expected. As the slope value increases, to minimize the least square fitting for the
majority of the data sample located around 0.1 - 1 mm hr$^{-1}$, $C_{(R=0.05)}$ must mathematically decrease.
As a further illustration, we performed the WRA fitting with a slope of 6 for the KaSACR
observations in Figure 5a. The fitted relation is plotted as the red dashed line in Figure 6. One finds
that the fitted Ze with slope of 6 lies between Frasier et al. (2013) and Gorgucci et al. (2013). For
most of the data samples (located around 0.1 - 1 mm hr$^{-1}$), the difference between the two WRA
fitting results is within 1 dB. The resulting $C_{(R=0.05)}$ with slope of 6 is larger than that with slope of
8.  However, the offset deviation due to possible fitting slope-fit change (Table 2) is 3 dB and
within the standard deviation of the estimated Ze as a function of $R$ (~3 dB). Thus, even with slope
fitting errors aassociated with this relative WRA technique, most drifts in the resulting long-term
calibration trend larger than the 3 dB would be meaningful and identifiable.

The calculated KAZR calibration offsets during the entire TRACER campaign are shown

in Fig. 8a (black asterisk for the daily value, thin dash line for the mean campaign-wide trend). We
find that the calibration offsets are relatively stable at around 2 dB with a standard deviation of 3
dB until 1 July 2022 (273 days since 1 Oct. 2021 in Fig. 12). After that time, the calibration offset
increases to around 7 dB in September. This late-period offset drift exceeds 5 dB, probably due to
deterioration of radar components in heavy rains in July and August. This shift is larger than the
uncertainty of the fitting method and the standard deviation of the fitting data.

**4.2 Evaluation of the TRACER KAZR Calibration Trend**

By monitoring the $Dze_{(R=0.05)}$ from every rainy day that meets our stratiform and duration

selection criteria, we determine a relative radar calibration offset trend. This offset has an
additional uncertainty associated with its fitting uncertainty and the assumption of negligible WRA
at $R \sim 0.05$ mm hr$^{-1}$. The combination of this WRA fitting technique with other typically less
frequent absolute radar calibration references would be ideal and cost-effective for the KAZR
long-term calibration. To evaluate the KAZR calibration offset trend during the entire TRACER
campaign, we performed three separate tests to demonstrate the potential offset uncertainty and/or
advantage of the current WRA fitting technique as compared to other established methods.



### *4.2.1 Direct KAZR-Disdrometer Comparison Near to Light Rain Onset*

As previously mentioned, a wet radome film may not form immediately at the onset of light rain, therefore the WRA is often assumed to be negligible when calibrating radar using disdrometer measurements near these rain onset windows. We perform a direct KAZR-disdrometer comparison at/near light rain onset in rain events for qualifying KAZR calibration events. The onset mean offset of each day is calculated if there are data samples with $R < 0.1$ mm hr$^{-1}$ lasting for 5 consecutive minutes from each observed rain event in the day. The onset mean offsets are shown in Fig. 8a (red diamonds). For the days with onset mean offset, these are typically close to the offsets from those calculated using the WRA fitting technique. However, the application of this method depends on the variation in precipitation rate over the 5-minute sampling period and the VDISQUANTS minimum sensitivity. The former causes large uncertainty and the latter causes fewer data samples, as shown in Fig. 8a.

### *4.2.2 WRA Fitting Technique Against the Calibrated RWP Ze*

As an independent cross-comparison, we also perform the WRA fitting technique with respect to calibrated RWP Ze at RWP time resolution (less than 8 s) with interpolated disdrometer rain rates over the entire TRACER campaign. Now the *Dze* is replaced with the difference between KAZR and RWP measurements. The WRA calibration offsets using the RWP measurements are shown with black asterisks on Fig. 8b. First, we notice that there are fewer RWP data points available. This is due to RWP mode switching in transient rain events. For the days that RWP measurements are available, the calibration offsets are very close to those derived using the disdrometer-estimated Ze in Fig. 8a and direct disdrometer checks therein. The drift in the offset trends from the start of July into September is smoother and clearer than that against the disdrometer measurement probably due to better temporal resolution. Overall, the calibration offset consistency in temporal trend and magnitude against the disdrometer and RWP measurements is an indicator of the good performance of the new WRA fitting technique.

### *4.2.3 Cross-Comparison Between KaSACR and KAZR*

As mentioned previously, KaSACR calibration offsets are stable between May and September of 2022. Furthermore, its calibration offsets calculated from the WRA fitting technique with the scanning VPT and stationary VPT measurements in Figure 6 are approximately -0.9 to -





0.1 dB, respectively, and 1dB from the direct disdrometer comparison at light rain onset. We
tentatively assign 0 dB calibration offset for KaSACR observations. Then cross-comparison
between KaSACR VPT mode and KAZR observations can also be used to quantify the KAZR
calibration offset trend. As KaSACR and KAZR operate at the same frequency, this cross-
comparison is done with full-profile samples rather than at certain height level since the cumulative
gaseous and rain attenuation should be same at each range gate.

For this cross-comparison, we first allocate the closest KaSACR profiles to KAZR profiles

and interpolate the KaSACR height range to the KAZR height range. Then, we select the data
sample using a signal-to-noise ratio threshold of 5 dB for both KaSACR and KAZR. In
precipitating events, the KaSACR in scanning VPT is expected to have a sawtooth or modulated
WRA cycling behavior, while the KAZR VPT is under a consistent/continuous WRA (see Fig. 2).
We screen the collocated profiles into precipitating and non-precipitating time periods using the
collocated surface rain rate from disdrometer measurements. Finally, the daily mean offsets
between KaSACR and KAZR observations in non-precipitating clouds are calculated and shown
in Fig. 8b (red diamonds). We find these calculated offsets have a very similar trend to those from
the WRA fitting technique against RWP measurement in Fig. 8b. This further supports the viability
of the WRA calibration offset behaviors and confidence in the offset drift we observed at the end
of the campaign.

**5 Summary**

In this study, we have demonstrated the wet radome influence on Ka-band radar

observations through comparisons that included KaSACR VPT observations under scanning (that
may shed water buildup) and stationary (non-shedding) conditions. The WRA is attributed to both
wet film and cumulative rainwater collecting on the radar radome. This attenuation influence
increases, as the rain rate increases. In campaign settings, it was found this attenuation may exceed
10 dB under a modest rain rate of 5 mm hr$^{-1}$. Taking advantage of the intrinsic WRA dependence
on rain rates, a new relative calibration monitoring technique was developed for use with the ARM
KAZR (or similar cloud radar systems) observations as obtained in moderate rain events from the
AMF1 deployment in Houston, TX during the TRACER field campaign.

The well-correlated relation between *Dze* and *R* (in logarithmic space) on precipitating

days is fitted with a log-linear equation, which has a similar tendency as the published WRA in



Frasier et al. (2013) and Gorgucci et al. (2013). This behavior serves as the basis for this relative WRA calibration technique. The corrected KAZR Ze with fitted *Dze,* which includes the WRA and Ze offset*,* agrees very well with both disdrometer-estimated and RWP-measured Ze. The radar calibration offset is calculated from the fitted *Dze* -*R* relation when *R* equals 0.05 mm hr $^{-1}$, assuming WRA is negligible at this light rain rate. The daily fitted slopes over the course of the TRACER campaign vary between 6 and 10 due to different data sampling in different rain types. A slope sensitivity study suggests that the calibration offset deviations due to slope variation are likely within the standard deviation of the estimated Ze as function of *R*, as well as those typical of underlying/collocated disdrometer measurement uncertainty (i.e. ~2-3 dB). The KAZR calibration offsets calculated with a constant slope of 8 during the TRACER campaign are stable near 2 dB compared to the disdrometer estimate with a standard deviation of 3 dB through June 2022. After that time, the calibration offsets increase to more than 7 dB.

The performance of the WRA fitting calibration technique is evaluated by comparing it with direct disdrometer measurements at the onset of rain events. The wet-radome technique consistently identifies a sound calibration offset over the entire project and arguably outperforms the direct disdrometer and radar comparison at the onset of light rain by reducing noise and increasing temporal consistency. The WRA fitting calibration technique is also applied to the KAZR observation against the calibrated RWP Ze reference. This test reveals sound performance and a clear and smooth matching trend in the July to September change in TRACER KAZR offsets, indicating that the new technique can be applicable to other calibrated reference radars with collocated surface rain rate measurements. The KAZR offset assessed from the cross-comparison between the stable and calibrated KaSACR VPT mode and KAZR observations in non-precipitating clouds also agree with the calibration offset trend from the WRA fitting technique. Moreover, determining the calibration offset and monitoring the long-term trend of ARM KAZR is the first step towards studying cloud seasonal and inter-seasonal variation. Having an easily adjustable cloud radar calibration method with collocated disdrometer or RWP data available will also facilitate cloud microphysical property retrieval, cloud process studies, and cloud variation associated with climate change using ARM KAZR measurements.

Since the technique may consider data samples collected during a wider range of light or moderate rain cases, it has a far less stringent requirement that other shorter-wavelength radar monitoring concepts using disdrometers or other radars that necessitate cloud, drizzle or light rain



observations at rain onset. One plan is to test whether this newly developed WRA technique may
be applicable to other cloud radars at ARM fixed sites (i.e., those in more/less humid, marine
and/or oceanic environments), or to what extent further site-specific refinement is needed for
different radar and sampling parameters. Recently, this WRA monitoring techinque has been
applied to measurements during other ARM field campaigns such as surface atmosphere integrated
field laboratory (SAIL) and eastern pacific cloud aerosol precipitation experiment (EPCAPE).
Along with TRACER, the resulted offset trends from those three campaigns are evaluated
favarably with the results from other KAZR calibration technique done independently in ARM
radar b1 data processing reports (Feng et al 2024, Matthew et al. 2024, Rocque et al 2024).
























Table 1. List of parameters for KAZR GE mode, KaSACR/XSACR in vertical pointing (VPT)
mode, and RWP in precipitation mode.

|  | KAZR (GE mode) | KaSACR (VPT mode) | XSACR (VPT mode) | RWP (Precipitation mode) |
|---|---|---|---|---|
| **Frequency (GHz)** | 34.0 | 35.3 | 9.71 | 1.29 |
| **Wavelength** | 8.57mm | 8.50mm | 3.09cm | 23.3cm |
| **Beam width (degree)** | 0.3 | 0.3 | 1.0 | >3 |
| **Time resolution (s)** | 2 | 4 | 3 | 5-8 |
| **Range resolution (m)** | 30 | 25 | 25 | 225 |
| **Minimum range (m)** | 160 | Others: 428 <br> 0903/04: 453 | 288 | 335 |
| **Radome diameter (m)** | 1.82 | 1.82 | 1.82 | N/A |




Table 2. Sensitivity study of the slope value in the log-linear fitting for KAZR and KaSACR
calibration on 03-04 September 2022 case in Figure 1. *b* and *a* are the slope and constant,
respectively, in the log-linear fitting in Eq. 2. $D_{Ze}(R=0.05)$ is the radar calibration offset when rain
rate ($R$) equals 0.05 mm hr $^{-1}$. More details can be found in Section 3.3.

| | KAZR | | | | KaSACR | | | |
|---|---|---|---|---|---|---|---|---|
| *b* | *a* | $D_{Ze}$ (*R*=0.05) | Correlation coefficient (rr) | Standard deviation (dB) | *a* | $D_{Ze}$ (*R*=0.05) | Correlation coefficient (rr) | Standard deviation (dB) |
| **6** | 17.1 | **9.3** | 0.88 | 3.8 | 9.8 | **2.0** | 0.89 | 3.4 |
| **8** | 18.1 | **7.7** | 0.90 | 3.9 | 10.9 | **0.5** | 0.91 | 3.4 |
| **8.6** | 18.5 | **7.3** | 0.91 | 4.1 | 11.1 | **-0.1** | 0.92 | 3.5 |
| **10** | 19.1 | **6.3** | 0.92 | 4.4 | 12.0 | **-1.0** | 0.93 | 3.7 |






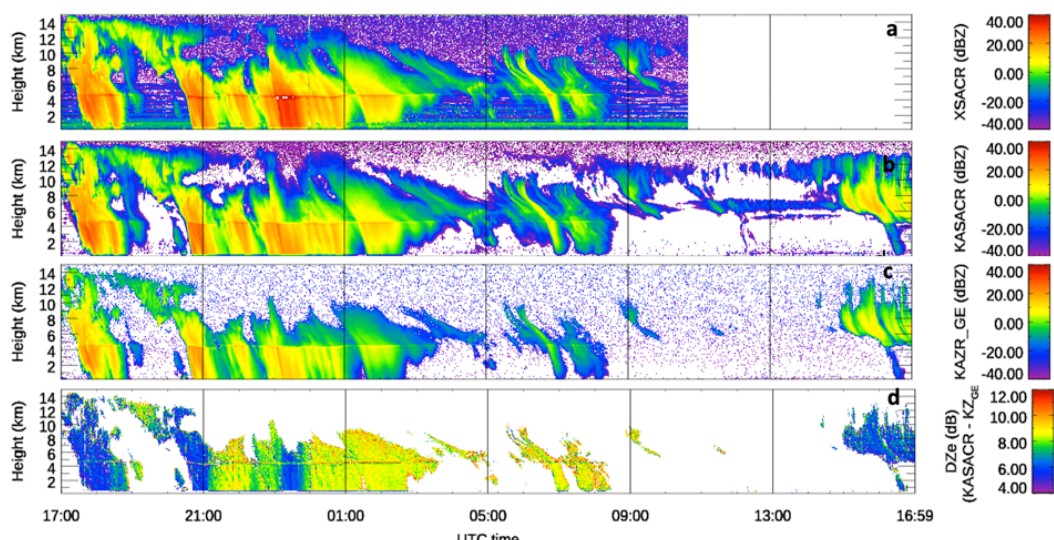

Figure 1. Measured radar reflectivity on 03-04 September 2022 from the TRACER field campaign.
a) XSACR, missing data after 10:40 UTC on 04 September 2022, b) KaSACR, c) KAZR GE mode,
d) Ze difference (DZe) between the KaSACR and the KAZR GE mode.



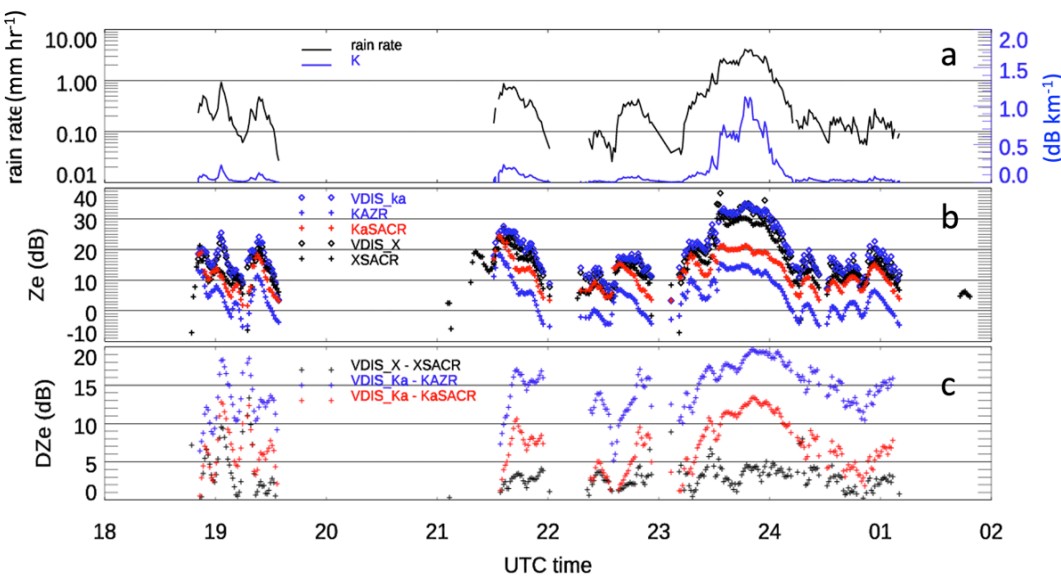

Figure 2. Measurements and comparison on 03-04 September 2022 between VDISQUANTS and radars. a) the timeseries of VDISQUANTS rain rate (black line) and rain droplet specific attenuation coefficients (*K*, blue line) at Ka band. b) the time series of measured Ze from KAZR GE (blue +), KaSACR (red +), and XSACR (black +) at 500 m after gaseous and rain attenuation corrections, and estimated Ze from VDISQUANTS at Ka (blue diamond) and X (black diamond) bands. c) Ze difference (DZe) between radar and disdrometer for XSACR (black cross), KaSACR (read cross), and KAZR (blue cross).  For this case, SACR was operated in the stationary VPT mode.




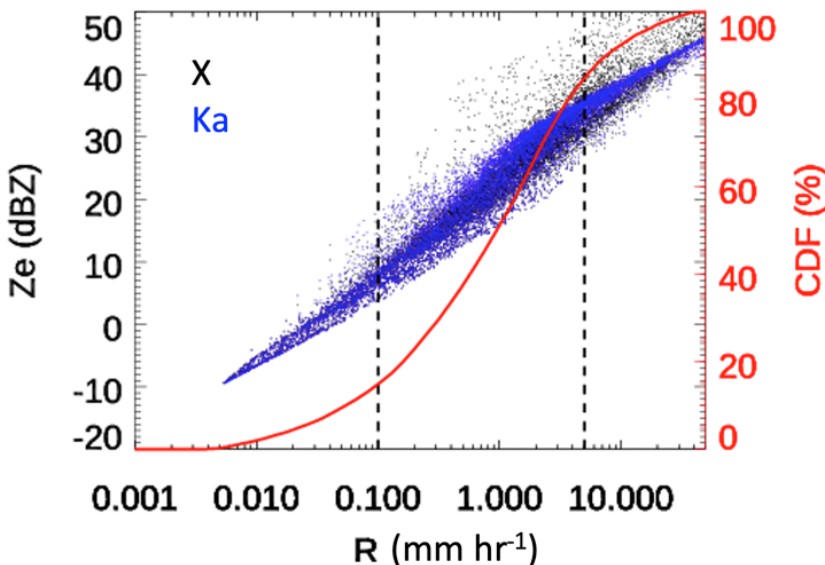

Figure 3. The estimated Ze from VDISQUANTS for Ka (blue dots) and X bands (black dots) during the entire TRACER campaign, plotted as a function of rain rate ($R$). The red line is the cumulative probability function (CDF) of $R$. The two vertical black lines are at rain rates of 0.1 and 5.0 mm hr$^{-1}$, respectively.



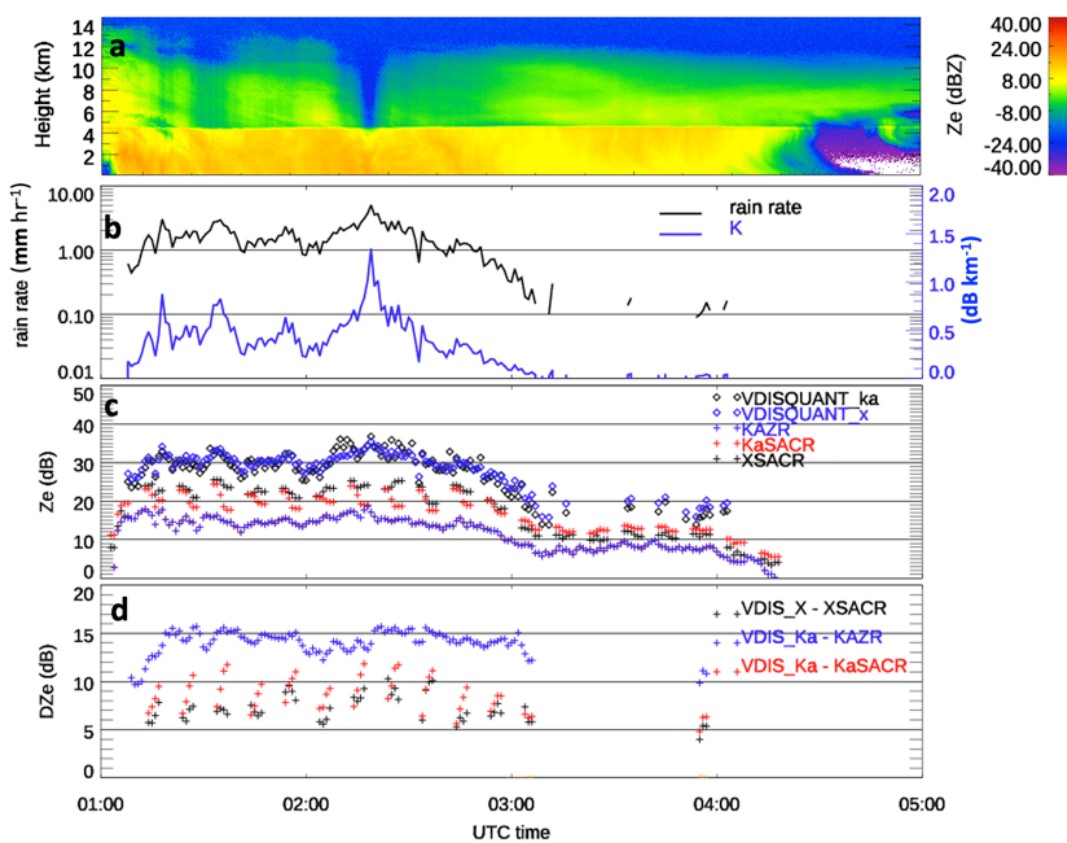

Figure 4. Radar and VDISQUANTS comparison for the case on August 11. a) Measured radar reflectivity (Ze) from the KAZR GE mode. b-d are similar to Fig. 2a-c. For this case, KaSACR and XSACR measurements are the scanning VPT mode and collocated with the VDISQUANTS with a ±2 minutes averaging window.





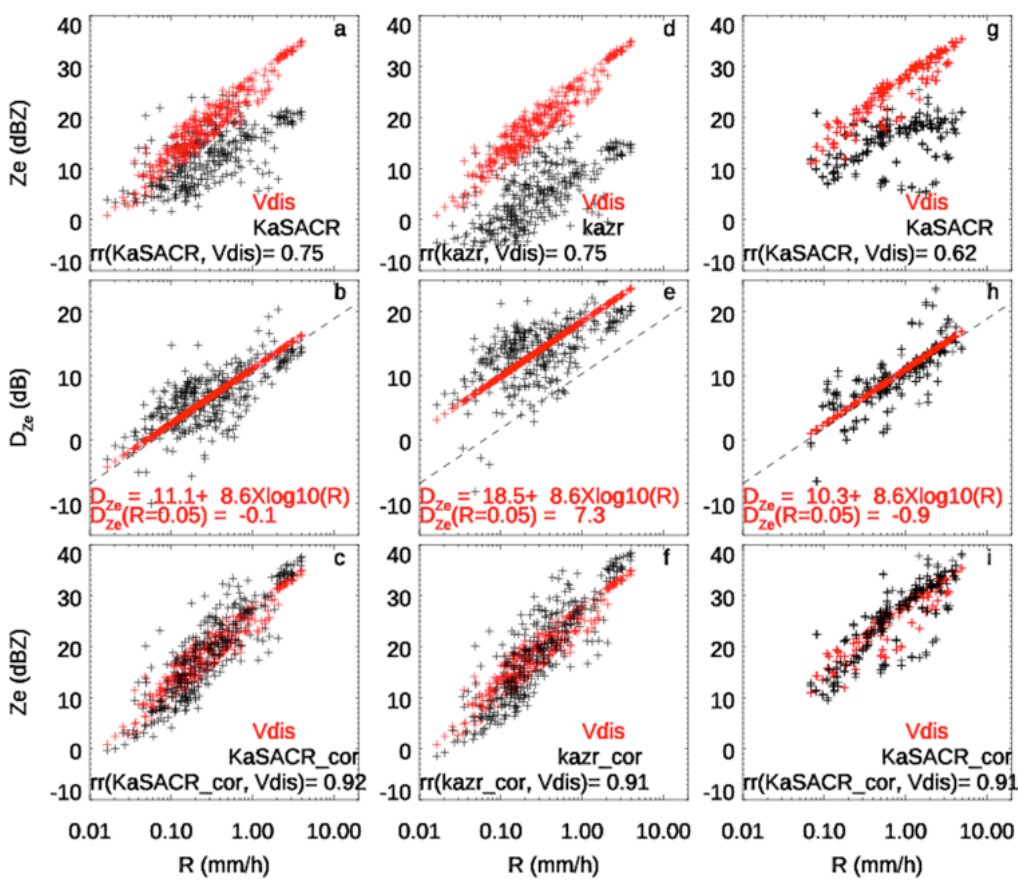

Figure 5. a) Scatter plot of radar measured Ze (black cross) at 500 m and VDISQUANTS-estimated Ze (red cross) as a function of rain rate $R$, b) Difference between measured Ze and VDISQUANTS-estimated Ze (Dze in black). The log-linear fitting in Eq.2 with slope $b$ at 8.6 are plotted in red cross,  c) Scatter plot of radar measured Ze (black cross) after log-linear fitting correction along with the VDISQUANTS-estimated Ze (red cross) for KaSACR stationary VPT (a-c) and KAZR GE (d-f) on 03-04 September, and KaSACR stationary VPT (g-i) collected on May 25, August 05, 11, 19 and 29. The correlation coefficients between the measured Ze and estimated Ze (rr) before and after the fitting correction are noted. The dashed black lines in second row (b, e, h) are the log-linear fitting with a= 10.3 and b= 8.6 for KaSACR in Table 2.







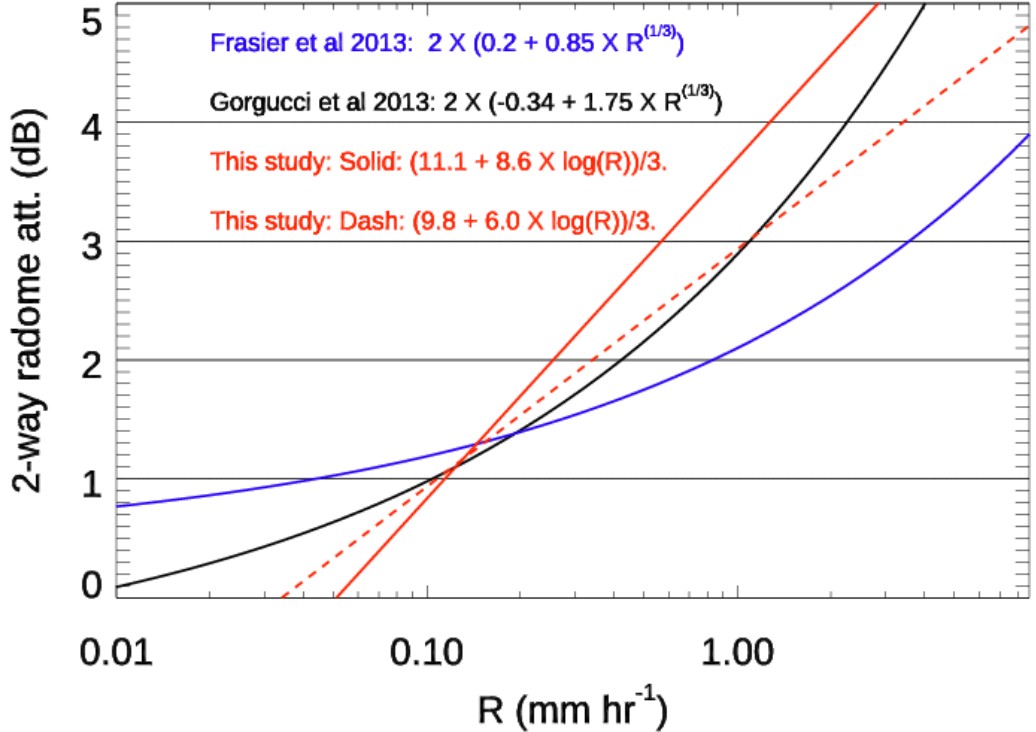


Figure 6.  Two-way radome attenuation as a function of rain rate ($R$) using the log-linear WRA
fitting relation in Eq. 2 with slopes of 8.6 (solid red) and 6.0 (dashed red) in this study at Ka-band,
which is divided by 3 and compared with two previous studies about X-band radars from Frasier
et al. 2013 and Gorgucci et al. 2013.










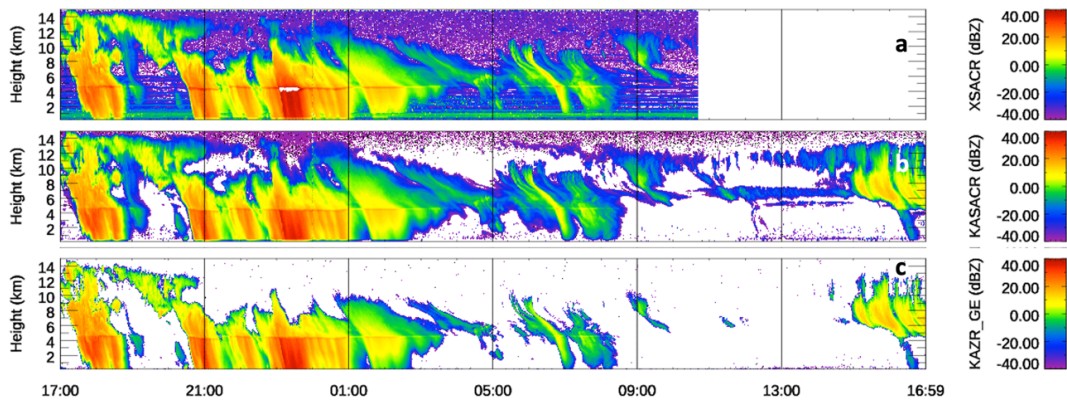


Figure 7. The same as Figure 1a-c except after WRA correction and radar calibration. For the
precipitating period, KaSACR is corrected using Eq. 2, with a slope of 8.6 and constant of 11.1.
XSACR is corrected with the offset of 3 dB from VDISQUANTs (black cross in Fig 2d), and
KAZR GE mode is corrected using Eq. 2, with a slope of 8.6 and constant of 18.5. For non-
precipitating periods, the calibration offsets of KaSACR and XSACR are assumed to be 0 dB,
while the KAZR GE mode is calibrated with offset of 7 dB.



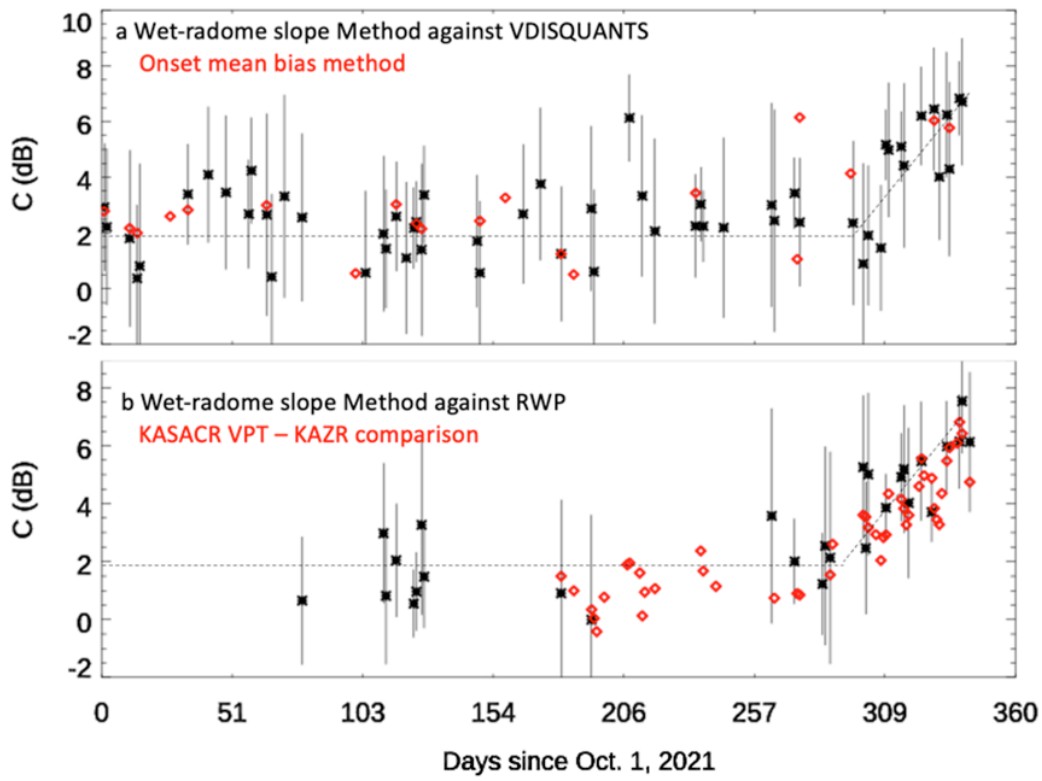

Figure 8. a) KAZR daily calibration offsets ($C$) from the mean KAZR bias method at the onset of light rain (red diamond) and the WRA fitting technique (black asterisk) against the VDISQUANTS data. Black vertical bar is the standard deviation of corrected Ze against the estimated Ze. b) KAZR daily calibration offset from the WRA fitting technique against the calibrated RWP measurement in black asterisk with vertical standard deviation bar. Red diamonds stand for the daily cross-comparison between the KaSACR VPT mode and the KAZR GE mode in non-precipitating clouds since May 26, 2022. The dashed black line is the mean trend outline from the WRA fitting technique in Fig. 8a.



**Data availability**

The KAZR, KaSACR and XSACR data at the TRACER campaign in this study are a1-level data. The surface disdrometer VDISQUANTS and interpolated sounding data are c1-level value added product data. They are all available at ARM data discovery at https://adc.arm.gov/discovery/#/ and through the following DOIs. The calibrated radar wind profiler data is ARM PI product and can be obtained from the data developer, Dr. Christopher R. Williams, through email (christopher.williams@colorado.edu) contact.

Bharadwaj, Nitin, Hardin, Joseph, Isom, Bradley, Johnson, Karen, Lindenmaier, Iosif, Matthews, Alyssa, Nelson, Danny, Feng, Ya-Chien, Deng, Min, Rocque, Marquette, Castro, Vagner, and Wendler, Tim. *Ka-Band Scanning ARM Cloud Radar*. United States: N. p., 2021. Web. doi:10.5439/1469302.

Bharadwaj, Nitin, Hardin, Joseph, Isom, Bradley, Johnson, Karen, Lindenmaier, Iosif, Matthews, Alyssa, Nelson, Danny, Feng, Ya-Chien, Deng, Min, Wendler, Tim, Castro, Vagner, and Rocque, Marquette. *X-Band Scanning ARM Cloud Radar*. United States: N. p., 2021. Web. doi:10.5439/1469303.

Hardin, Joseph, Giangrande, Scott, and Zhou, Aifang. *ldquants*. United States: N. p., 2019. Web. doi:10.5439/1432694.

Hardin, Joesph, Giangrande, Scott, Fairless, Tami, and Zhou, Aifang. *vdisquants: Video Distrometer derived radar equivalent quantities. Retrievals from the VDIS instrument providing radar equivalent quantities, including dual polarization radar quantities (e.g., Z, Differential Reflectivity ZDR)*. United States: N. p., 2021. Web. doi:10.5439/1592683.

Isom, Bradley, Nelson, Danny, Andrei, Iosif, Hardin, Joseph, Matthews, Alyssa, Johnson, Karen, Bharadwaj, Nitin, Feng, Ya-Chien, Rocque, Marquette, Deng, Min, Wendler, Tim, and Castro, Vagner. *ARM: KAZRCFRGE*. United States: N. p., 2018. Web. doi:10.5439/1498936.

Isom, Bradley, Nelson, Danny, Andrei, Iosif, Hardin, Joseph, Matthews, Alyssa, Johnson, Karen, Bharadwaj, Nitin, Feng, Ya-Chien, Rocque, Marquette, Deng, Min, Wendler, Tim, and Castro, Vagner. *ARM: KAZRCFRMD*. United States: N. p., 2018. Web. doi:10.5439/1498948.



Jensen, Michael, Giangrande, Scott, Fairless, Tami, and Zhou, Aifang. *interpolatedsonde*. United
States: N. p., 1998. Web. doi:10.5439/1095316.




**Author contribution**


MD developed the main idea of WRA calibration techiqnue. SG, MJ, KJ provided inputs on the

data analysis process.  CW provided the calibrated RWP and write-up of RWP data. JC,

YF, AM, MR and MD are the ARM radar data mentor team. They provided TRACER

related radar information and additional KAZR calibration as shown in TRACER b1 data

processing. IL and TW are the ARM radar engineers, providing important information on

radar hardware and software and radar saturation information. AZ and DW are the

disdrometer mentor and VAP developer. ZZ and EL provided inputs on radar wet radome

attenuation in this study. All coauthors helped to edit and comment the manuscript draft.



**Competing interests**
The authors declare that they have no conflict of interest.





**Acknowledgement**
We acknowledge the exceptional work of the radar engineering team and data mentor team for the
close to 100% operation rate of KAZR during the TRACER campaign. We would like to thank
the ARM TRACER team for the quality data of KaSACR, XSACR, disdrometer, RWP and
interpolated sounding measurements. Contributions from Brookhaven National Laboratory co-
authors were supported by the Atmospheric Radiation Measurement (ARM) Facility and the
Atmospheric System Research (ASR) program of the Office of Biological and Environmental
Research in the U. S. Department of Energy, Office of Science, through Contract No. DE-
SC0012704.  Dr. C.R. Williams and the RWP work is supported under ASR grant number DE-
SC0021345. Pacific Northwest National Laboratory (PNNL) is operated by Battelle for the U. S.
Department of Energ. The authors from PNNL are also supported by ARM through Contract
No. DE-SC0015990.



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
