# Peer review of "Wet-Radome Attenuation in ARM Cloud Radars and Its Utilization in Radar Calibration Using Disdrometer Measurements Min Deng1, Scott E. Giangrande1, Michael P. Jensen1, Karen Johnson1, Christopher R. Williams2, Jennifer M. Comstock3, Y"

_EGUsphere, 2024_

## Author Comment (AC1)

This manuscript describes an approach to identify absolute calibration trends of cloud radars. It is an important topic since calibration offsets cause biases in retrievals which use radar reflectivity measurements. I believe that revisions are necessary before the publication of the manuscript.

Thank you for your feedback and for emphasizing the importance of addressing calibration trends in cloud radars to reduce biases in radar-based retrievals. We appreciate your suggestion for revisions and are committed to enhancing the clarity and impact of the manuscript. The response to your comments and suggestion are provided in the following in blue.

Main comments.

1. The fitting calibration technique requires the use of Dze values, which are the differences between measured and disdrometer-based reflectivities with subsequent estimation of the intercept of Dze-log(R) fits. The authors need to clarify better why the calibration offset estimated directly by comparing radar measurements when R is small (e.g., at the onset of precipitation, so WRA is negligible) is not good enough.

This related discussion is in line 163-172 in the introduction and line 310-314 in Section 3.1, Section 4.2.1, and in the summary.

The direct comparison concepts previously cited typically consider only the periphery cloud, drizzle or light rain conditions (i.e., $R < 1\text{-}2$ mm hr$^{-1}$) at the onset of a rainfall event to minimize various forms of attenuation. This often is a very stringent and subjective employment of these conditions: First, it limits the opportunities for direct disdrometer monitoring of cloud radar to a selected window of rainfall rates and event timing. Identifying these light rain or drizzling conditions is also contingent on the requirements for collecting high-quality disdrometer measurements (i.e., those that require significant droplet number counts), wherein a separate rain rate cut-off may be required to avoid significant WRA. Its application during the TRACER campaign is shown in Figure 8. This approach provides results comparable to those obtained from the fitting calibration technique.

However, in cases of frequent deep convection during TRACER, the direct comparison becomes limited, resulting a reduction in usable data. The rainfall rate PDF in Figure 3 indicates that only 15% of all rainy dataset are below 0.1 mm/hr, even less at the onset of rain events, making the application of direct comparison in Figure 8 less frequent compared to the fitting calibration technique.

2       The KAZR radome is tilted by 4 deg (line 184) to remove the rain water and minimize wet-radome effects. Did this design prove to be not effective? Would your results hold for different tilted angles (e.g., reducing WRA magnitude with the tilt increase)?

The radome tilt at a 4-degree angle does indeed help expedite drying after a rain event, reducing the duration of WRA. However, during active rain, a layer of water can still form on the tilted radome — similar to the rain sheet on a car window — resulting in some attenuation despite the tilt. While exploring the sensitivity of WRA to various tilt angles would indeed be interesting, a detailed analysis of how increased tilt might reduce WRA falls outside the scope of this study.

3.      WRA are likely dependent on the radome type, because different radomes remove rain water with different efficiency. Given this, one would expect different slopes of Dze-log(R) relations for KaSACR and KAZR. You assume that these slopes are the same (~8.6 in Figs.5). Please clarify/explain.

We agree that WRA can vary across different radars, primarily due to differences in radome type, age, and water-removal efficiency.

The comparison of KAZR and KaSACR performance observed on September 3-4, 2022 (Figure 1) show the WRA sensitivity to difference radomes and its aging. The difference between KAZR and KaSACR radars (in GE mode) minimizes in measurements for both high cirrus clouds and moderate rain. Specifically, the minimum difference in high cirrus clouds suggests that both radars perform similarly in these conditions, with the difference likely due to a calibration offset. During moderate rain, minimum difference indicates that WRA for both KAZR and KaSACR behaves similarly and is primarily influenced by rain rate. In light rain, however, the KAZR is 1-2 dB lower than KaSACR, and this bias persisted longer after the rain had stopped (Figure 1d). This suggests that WRA effect in the KAZR is slightly stronger and more prolonged likely due to its older radome, which has reduced hydrophobic efficiency. The following figure shows the one-to-one comparison of KAZR and KaSACR Ze at 500 m, which illustrates the well correlated relation with about 7 dB mean bias and the 1-2 dB difference in light and moderate rain.

[Figure]

Despite this 1-2 dB difference in light rain, they do not introduce significant biases in the calibration offset calculation when using a wide range of WRA data in our fitting technique, probably that the 1-2 dB WRA difference is not significant compared to the WRA dependence on rain-rate, ~8 dB over rain rates from 0.1 to 1 mm/hr and the associated deviation due to data collocation.

Therefore, the slope of 8.6 seems fit the data collected for KAZR and KaSACR under scanning or stationary VPT modes in Figure 5. However, for cases with only light rain or heavy rain, the fitting may be biased due to the slope difference. To decrease the uncertainty in the WRA fitting technique, we can increase the fitting window from 1 day to 5 days to increase the data sample. In Figure 9, we smoothed the calibration offsets with 2-day window to illustrate that large window for WRA fitting could decrease the random deviation associated with fitting uncertainty.

4.      What is the reason of the large change in the KAZR calibration offset (up to ~7 dB or so) towards the end of the deployment (Fig. 8)?

The large change in the KAZR calibration offset toward the end of the deployment, is found to related to a drop of about 1 dB in transmitter power toward the project's conclusion. This issue has been documented in the TRACER radar b1-data processing, as shown in the following figure. Now this discussion is added in Section 4.1.

[Figure]

5       It appears that for R > 2 mm/h, your log-linear WRA- log(R) relation deviates significantly from the R^(1/3) behaivior suggested by previous studies (Fig. 6).

The log-linear fit diverges from the R^(1/3)  behavior reported by Gorgucci et al. (2013) and Frasier et al. (2013) by around 1 dB when the rain rate exceeds 2 mm/hr or falls

below 0.1 mm/hr. However, the majority of the selected rain observations for our analysis lie within the 0.1-1 mm/hr range, where the fit demonstrates strong correlation and minimal bias. This 1 dB deviation is relatively minor compared to the broader range of WRA values, which span from 0 to 15 dB. Additionally, we observed discrepancies of up to 3 dB between KAZR and disdrometer measurements due to collocation differences, which predefines with the uncertainty of 3 dB inherent in the WRA fitting technique.

To test the sensitivity of calibration offset to fitting function, we applied the R^(1/3) behavior and compared the results with the linear fitting result in Figure 9, which is now added in the revised manuscript. We can see the R^(1/3) behavior and the loglinear fitting has a similar calibration offset trend.

[Figure]

6      Provide more information on frequency dependence of water absorption on a radome (line 142) as opposed to attenuation by cloud and precipitation drops, which is not scaled as the wavelength. The Bertie et al. 1996 reference (line 142) is not in the reference list.

Bertie et al. 1996 reference is now added to the reference list: John E. Bertie and Zhida Lan, "Infrared Intensities of Liquids XX: The Intensity of the OH Stretching Band of Liquid Water Revisited, and the Best Current Values of the Optical Constants of $H_2O(l)$ at 25°C between 15,000 and 1 cm$^{-1}$," Appl. Spectrosc. 50, 1047-1057 (1996)

The frequency dependence of water absorption is further illustrated with the following figure from D. J. Segelstein, "The complex refractive index of water," University of Missouri-Kansas City, (1981). The water absorption is inversely proportional to wavelength, therefore the water absorption ratio between X and Ka band is about 3.0cm/8mm =3~4.

[Figure]

7       Please add some discussion on applicability of your approach to different radars, different radomes and different disdrometers (e.g., PARSIVEL, Joss…).

The application of this WRA fitting technique depends on the uncertainty disdrometer measurements, the radome type and age. To be more conclusive, we also performed the WRA fitting technique against LDISQUANTS. The resulted calibration offsets are compared with those against VDISQANTS in Figure 9. We can see the calibration offsets are sensitive to different disdrometer measurements, however the general trends are very

similar that the mean calibration offset is around 2dB before July 2022, after that the calibration offset increases to about 7 dB.

Minor comments

a.                 Providing Ze – R relations for X and Ka-bands (using reflectivity as independent variable in such relations) in Fig. 3 would be informative.

Now Figure 3 is revised so that the Ze is the x axis as the independent variable in the Ze-R relation.

b.                 I believe you are assuming that rain-rate and DSDs at the disdrometer level and at the radar resolution volume are the same. Please specify.

To compare KAZR Ze at 500 m with disdrometer estimates at the surface, we assume that both observe the same rain. Therefore, the reflectivity should be similar after accounting for gas and rain attenuation, if WRA and calibration offsets for KAZR are zero. WRA occurs at the radar radome level, while the rain rate from the disdrometer is measured at the surface. For rain attenuation correction in KAZR, we assume a uniform rain layer between the surface and 500 m.

c.                 In different parts of the manuscript you use "dB" units instead of "dBZ" units (e.g., lines 216-217, Y-axis in Figs, 2b, 4c, and other instances), and "dBZ" units instead of "dB" units (e.g., lines 253, 271 and other instances).

Thank you for the careful reviewing. Now it is revised as dBZ unit for Ze, while Dze in 'dB'.

d.                 The manuscript could benefit from additional editing.

Thank you for the comment. A comprehensive edit of the manuscript has been completed to improve clarity and readability.

e.                 You use abbreviations "KaSACR" and "SACR" interchangeably. I suggest using KaSACR everywhere.

Agree, Thank you. Now it is revised accordingly.

---

## Author Response (AR1)

Response to Review 1

This manuscript describes an approach to identify absolute calibration trends of cloud radars. It is an important topic since calibration offsets cause biases in retrievals which use radar reflectivity measurements. I believe that revisions are necessary before the publication of the manuscript.

We greatly appreciate your valuable feedback and for emphasizing the importance of addressing calibration trends in cloud radars to reduce biases in radar-based retrievals. Your suggestions are insightful, and we are committed to enhancing the clarity and impact of our manuscript. Below, we provide responses to your comments and suggestions in blue.

Main comments.

1. The fitting calibration technique requires the use of Dze values, which are the differences between measured and disdrometer-based reflectivities with subsequent estimation of the intercept of Dze-log(R) fits. The authors need to clarify better why the calibration offset estimated directly by comparing radar measurements when R is small (e.g., at the onset of precipitation, so WRA is negligible) is not good enough.

This related discussion is included in lines 163–172 of the introduction and in lines 310–314 of Section 3.1, Section 4.2.1, and the summary.

The direct comparison method you mentioned focuses on small rainfall rates (e.g., drizzle or light rain, R < 1–2 mm/hr) at the onset of precipitation to minimize attenuation effects. While this is a valuable approach, it presents limitations:

- Data availability: It restricts analysis to a narrow window of rainfall rates and event timings, reducing the opportunities for monitoring.

- Disdrometer requirements: High-quality disdrometer measurements require sufficient droplet counts, necessitating an additional rain-rate cutoff to avoid significant wet radome attenuation (WRA).

Figure 8 illustrates the application of this direct comparison method during the TRACER campaign, showing results comparable to those obtained using the fitting calibration technique. However, during frequent deep convection events in TRACER, this approach results in limited usable data, as the rainfall rate PDF in Figure 3 reveals that only 15% of the rainy dataset corresponds to R < 0.1 mm/hr, with even fewer observations at the onset of precipitation. Thus, the direct comparison method is less applicable than the fitting technique.

.

2       The KAZR radome is tilted by 4 deg (line 184) to remove the rain water and minimize wet-radome effects. Did this design prove to be not effective? Would your results hold for different tilted angles (e.g., reducing WRA magnitude with the tilt increase)?

The radome tilt at a 4-degree angle does indeed help expedite drying after a rain event, reducing the duration of WRA, as shown in Figure 1. However, during active rain, a layer of water can still form on the tilted radome — similar to the rain sheet on a car window — resulting in some attenuation despite the tilt. While exploring the sensitivity of WRA to various tilt angles would indeed be interesting, a detailed analysis of how increased tilt might reduce WRA falls outside the scope of this study.

3.      WRA are likely dependent on the radome type, because different radomes remove rain water with different efficiency. Given this, one would expect different slopes of Dze-log(R) relations for KaSACR and KAZR. You assume that these slopes are the same (~8.6 in Figs.5). Please clarify/explain.

We agree that radome type and aging can significantly influence WRA. Our comparison of KAZR and KaSACR (September 3–4, 2022; Figure 1) highlights the differences:

- Cirrus clouds: Minimal differences suggest calibration offsets are the primary factor.
- Moderate rain: Both radars show similar WRA behavior, predominantly influenced by rain rate.
- Light rain or after rain: KAZR shows a 1–2 dB lower response than KaSACR, likely due to reduced hydrophobic efficiency in its older radome.

Despite these differences, they do not introduce significant biases in the calibration offset calculation with daily measurement, as the 1–2 dB variation is negligible compared to the broader ~7 dB WRA dependence on rain rate in the following figure. To mitigate potential biases, we increased the fitting window with 2day smoothing, as illustrated in Figure 9, which shows that larger fitting windows reduce random deviations and improve reliability.

[Figure]

Figure 1 the one-to-one comparison of KaSACR and KAZR at 500 m for September 2-3.

4. What is the reason of the large change in the KAZR calibration offset (up to ~7 dB or so) towards the end of the deployment (Fig. 8)?

The significant (~7 dB) calibration offset increase toward the project's conclusion corresponds to a 1 dB drop in transmitter power, as documented in TRACER radar b1-data processing and in the added figure 9 in the following. This discussion has been added to Section 4.1.

5 It appears that for R > 2 mm/h, your log-linear WRA- log(R) relation deviates significantly from the R^(1/3) behavior suggested by previous studies (Fig. 6).

The log-linear fit diverges from the R^(1/3) behavior reported by Gorgucci et al. (2013) and Frasier et al. (2013) by around 1 dB when the rain rate exceeds 2 mm/hr or falls below 0.1 mm/hr. However, the majority of the selected rain observations for our analysis lie within the 0.1 - 1 mm/hr range, where the fit demonstrates strong correlation and minimal bias. This 1 dB deviation is relatively minor compared to the broader range of WRA values, which span from 0 to 15 dB. Additionally, we observed discrepancies of up to 3 dB between KAZR and disdrometer measurements due to collocation differences, which predefines with the uncertainty of 3 dB inherent in the WRA fitting technique.

To test the sensitivity of calibration offset to fitting function, we applied the R^(1/3) behavior and compared the results with the linear fitting result in Figure 9, which is now added in the revised manuscript. We can see the R^(1/3) behavior and the loglinear fitting has a similar calibration offset trend.

[Figure]

Figure 9 KAZR daily calibration offsets ($C$) from log-linear fitting with Eq. 2 (red asterisk with black standard deviation bar) or the $R^{1/3}$ relation (blue diamond) against a) LDQUANTS and b) VDISQUANTS data. The daily offsets are smoothed with 2-day window. c) KAZR transmitted power. Noticeable decrease of transmitted power is well correlated with the increase of calibration offset.

6    Provide more information on frequency dependence of water absorption on a radome (line 142) as opposed to attenuation by cloud and precipitation drops, which is not scaled as the wavelength. The Bertie et al. 1996 reference (line 142) is not in the reference list.

Bertie et al. 1996 reference is now added to the reference list: John E. Bertie and Zhida Lan, "Infrared Intensities of Liquids: The Intensity of the OH Stretching Band of Liquid

Water Revisited, and the Best Current Values of the Optical Constants of $H_2O(l)$ at 25°C between 15,000 and 1 cm$^{-1}$," Appl. Spectrosc. 50, 1047-1057 (1996)

The frequency dependence of water absorption is further illustrated with the following figure from D. J. Segelstein, "The complex refractive index of water," University of Missouri-Kansas City, (1981). The water absorption is inversely proportional to wavelength, therefore the water absorption ratio between X and Ka band is about 3.0cm/8mm =3~4.

[Figure]

7       Please add some discussion on applicability of your approach to different radars, different radomes and different disdrometers (e.g., PARSIVEL, Joss…).

The application of this WRA fitting technique depends on the uncertainty disdrometer measurements, the radome type and age. To be more conclusive, we also performed the WRA fitting technique against LDQUANTS. The resulted calibration offsets are compared with those against VDISQANTS in Figure 9. We can see the calibration offsets are sensitive to different disdrometer measurements, however the general trends are very similar that the mean calibration offset is around 2dB before July 2022, after that the calibration offset increases to about 7 dB.

Minor comments

a.          Providing Ze – R relations for X and Ka-bands (using reflectivity as independent variable in such relations) in Fig. 3 would be informative.

Now Figure 3 is revised so that the Ze is the x axis as the independent variable in the Ze-R relation.

b.          I believe you are assuming that rain-rate and DSDs at the disdrometer level and at the radar resolution volume are the same. Please specify.

To compare KAZR Ze at 500 m with disdrometer estimates at the surface, we assume that both observe the same rain. Therefore, the reflectivity should be similar after accounting for gas and rain attenuation, if WRA and calibration offsets for KAZR are zero. WRA occurs at the radar radome level, while the rain rate from the disdrometer is measured at the surface. For rain attenuation correction in KAZR, we assume a uniform rain layer between the surface and 500 m.

c.          In different parts of the manuscript you use "dB" units instead of "dBZ" units (e.g., lines 216-217, Y-axis in Figs, 2b, 4c, and other instances), and "dBZ" units instead of "dB" units (e.g., lines 253, 271 and other instances).

Thank you for the careful reviewing. Now it is revised as dBZ unit for Ze, while Dze in 'dB'.

d.          The manuscript could benefit from additional editing.

Thank you for the comment. A comprehensive edit of the manuscript has been completed to improve clarity and readability.

e.          You use abbreviations "KaSACR" and "SACR" interchangeably. I suggest using KaSACR everywhere.

Agree, Thank you. Now it is revised accordingly.

The manuscript by Deng et al. describes a method for monitoring trends in calibration of ARM cloud radars and the effect of wet radome attenuation (WRA). Both aspects are important because the number of cloud radars working at Ka band is increasing also because there are (at some extent), affordable radars commercially available that are used mainly for research purposes. Their performance should be monitored and the dependency of the impact on measured reflectivity factor of the WRA, which seems predominant over precipitation and gas attenuation must be predicted. The proposed method is based on a linear relation between the difference (in dB) between the Ze predicted from disdrometer measurements at ground and the one radar measured by radar at 500 meters. With proper averaging, intercept and slope of the relation express the system bias and the rainfall-dependent WRA, respectively.

The current manuscript focuses on one ARM setup. It would be beneficial to generalize the method for different radars and disdrometers. Extending the method to laser disdrometers should not be too difficult, although laser disdrometer measurements are generally considered less accurate than video disdrometer measurements. Laser disdrometers should be available in the TRACER campaign. Extending the method to other types of radomes could be challenging, as radome attenuation depends on many factors, including the hydrophobicity of the radome material, its aging, and the geometry of the radome. The Gibbs formula and the works by Gorgucci et al., Frasier et al., and Schneebeli et al. (doi:10.5194/amt-5-2183-2012) refer to an X-band radar with a hemispherical radome, which might explain the different behavior observed in Figure 6.

Thank you for your thoughtful comments and suggestions. We appreciate your input on generalizing our method to different disdrometer measures and fitting functions.

1. **Generalization to Other Radars and Disdrometers**: We agree that extending the method to include different radar and disdrometer setups could broaden the applicability of our work. We will discuss in the revised manuscript with calibration offset calculation against both LDQUANTS and VDISQUANTS.

   **Explanation of Figure 6 Behavior**: We appreciate your observation regarding Figure 6 and the possible explanation for the differences based on radome characteristics. In the revision, we will expand our discussion to address how the WRA fitting technique is sensitive to the fitting function of loglinear and $R^{1/3}$ relations and also to the LDQUANTS and VDISQUANTS measurement. The following results are added in Figure 9.

   The daily calibration offsets show slight variation between LDQUANTS and VDISQUANTS, indicating minor differences in disdrometer measurements. While the

calibration offsets from the log-linear and $R^{1/3}$ fittings can differ by up to 2 dB, the overall trends remain similar, with a mean offset of approximately 2 dB before July 2022, increasing to around 7 dB afterward.

[Figure]

Figure 9 KAZR daily calibration offsets (*C*) from log-linear fitting with Eq. 2 (red asterisk with black standard deviation bar) or the $R^{1/3}$ relation (blue diamond) against a) LDQUANTS and b) VDISQUANTS data. The daily offsets are smoothed with 2-day window. c) KAZR transmitted power. Noticeable decrease of transmitted power is well correlated with the increase of calibration offset.

**Minor Suggestions:**

- Please replace "K" or "$K$" with "$k$" for specific attenuation to allow a better identification (e.g., in Figures 2a and 4a).

  It is revised accordingly.

- In Figure 5, please remove the "corr" within the panels or improve the panels.

  Now it is revised accordingly.

- The different uses of Ze, which can be predicted or measured with different radars, can be confusing. Consider using superscripts to differentiate them, although this is just a suggestion.

  Now the subscripts are added to differentiate them.

---

## Author Response (AR2)

Reviewer 1

This is a suitable paper describing an approach for monitoring radar calibration offsets. The overall author response to the comments was satisfactory. The general utility of the quantitative results of the study, however, could be quite limited because they are radome and radar dependent.

Thank you very much for your insightful review. We completely agree with your observation that wet-radome attenuation is dependent on both the radome and radar characteristics. To address this, we have incorporated the following statement in the final discussion:

> Future plans include testing this newly developed WRA technique at other ARM fixed sites (e.g., in more humid, marine, or oceanic environments) to assess the extent of any necessary site-specific refinements for different radars and sampling conditions, acknowledging that wet-radome attenuation is inherently dependent on both the radome and radar characteristics.

Minor comments
Please provide Z-R best fits for both X and Ka bands in Fig.3. Also, color separation between blue (Ka-band) and black (X-band) dots is pure. Consider changing colors.

Revised as suggested.

Line 178: change "dBZ" to "dB"

Revised as suggested.

Reviewer 2

The changes from the original manuscript have been noticed and result in much better manuscript.

I must emphasize a comment from the open review regarding the $R^{1/3}$ dependency. My previous comment was likely unclear. Of the two radars used in the study, the KAZR has a flat radome inclined at 4°, and the XSACR also has a flat radome inclined at 4°. The Gibbs formula with its $R^{1/3}$ dependency refers to a spherical radome. The cited works adopting this dependency (Gorgucci et al., Frasier et al., Bechini et al., and Kurri and Huuskonen 2008, Fig 1, Eq 6) refer to spherical radomes. Therefore, the radomes used in these papers are

spherical, which is a different geometry from the radomes adopted by KAZR and XSACR. I suspect that the different geometry could explain the variable performance of the fitting in Fig. 6.

However, I do not require the authors to verify this. Instead, I strongly recommend emphasizing in the manuscript, between lines 120 and 135, that the cited studies refer to systems using spherical radomes, which differ from the geometries of radomes adopted in KAZR and XSACR.

Thank you very much for this insightful comment. We have emphasized that the cited studies pertain to systems utilizing spherical radomes, which differ from the geometries of radomes used in KAZR and XSACR, as clarified in lines 120–135 of the manuscript. Additionally, we highlighted these differences in lines 378–380 when the WRA relationships are compared. Furthermore, we acknowledged in lines 597–598 that wet-radome attenuation is dependent on both the radome and radar characteristics